# Biomarkers of Intestinal Permeability Are Influenced by Diet in Patients with Ulcerative Colitis—An Exploratory Study

**DOI:** 10.3390/diagnostics14232629

**Published:** 2024-11-22

**Authors:** Natasha Haskey, Maximillian Eisele, Andreina Bruno, Raylene A. Reimer, Munazza Yousuf, Lorian M. Taylor, Remo Panaccione, Subrata Ghosh, Maitreyi Raman

**Affiliations:** 1Department of Medicine, Cumming School of Medicine, University of Calgary, 2500 University Drive NW, Calgary, AB T2N 1N4, Canada; natasha.haskey@ubc.ca (N.H.); maximilian.eisele@ucalgary.ca (M.E.); munazza.yousuf1@ucalgary.ca (M.Y.); lorian.taylor@lyfemd.com (L.M.T.); rpanacci@ucalgary.ca (R.P.); 2Department of Biology, Irving K Barber Faculty of Science, University of British Columbia-Okanagan, 3137 University Way, Kelowna, BC V1V 1V7, Canada; 3Institute of Translational Pharmacology, National Research Council of Italy (CNR), Via Ugo La Malfa, 153, 90146 Palermo, Italy; andreina.bruno@ift.cnr.it; 4Faculty of Kinesiology, University of Calgary, 2500 University Drive NW, Calgary, AB T2N 1N4, Canada; reimer@ucalgary.ca; 5APC Microbiome Ireland, College of Medicine and Health, University College Cork, T12 K8AF Cork, Ireland; sughosh@ymail.com

**Keywords:** ulcerative colitis, intestinal permeability, diet

## Abstract

**Background and Objectives:** The disruption of the intestinal epithelial barrier leads to increased intestinal permeability (IP), allowing endotoxins and pathogens to enter the bloodstream contributing to chronic inflammation. Western diets are associated with increased IP, while diets rich in polyphenols, fiber, and omega-3 fats are linked to decreased IP. The relationship between diet, disease activity, and IP in ulcerative colitis (UC) is poorly understood. We evaluated the effects of serum zonulin and lipopolysaccharide-binding protein (LBP) and their relationship to dietary factors in UC. **Methods:** A cross-sectional analysis was conducted on 37 UC participants who had baseline measures of dietary intake, disease activity and serum. Serum LBP and zonulin levels were measured by ELISA. Spearman’s rank correlations explored relationships between baseline IP, diet, and disease activity. **Results:** The median age was 35 years (29–47 years), 54% were male, and 76% were in clinical remission or had mild disease activity (partial Mayo score ≤ 4). LBP was significantly correlated (*p* < 0.05) with disease activity (partial Mayo score (r = 0.31), and positively correlated with total fat (r = 0.42) and refined grains (r = 0.35), but negatively correlated with fruit consumption (r = −0.50) and diet quality (r = −0.33). Zonulin was negatively correlated with protein (r = −0.39), niacin (r = −0.52) and vitamin B12 (r = −0.45) with a trend for significance (*p* = 0.06) with body mass index (r = 0.32). **Conclusions:** Baseline LBP levels were correlated with disease activity markers and dietary factors, suggesting that it could be a useful biomarker for assessing disease activity and diet quality in UC. Further trials are needed to confirm these findings.

## 1. Introduction

Inflammatory bowel diseases (IBDs), which include Crohn’s disease (CD) and ulcerative colitis (UC), are chronic inflammatory conditions of the gastrointestinal tract that often require several pharmacological approaches for inducing and maintaining remission. Despite advances in medical therapies, up to 18% of patients with CD and 7% with UC may still require surgery at some point in their lives [1]. While the introduction of new medications for IBD has been promising, a ‘therapeutic ceiling’ has emerged, with no single treatment option proving effective, with between 30–40% of patients not responding to biologic agents [2,3]. This limitation has driven a growing interest in using non-pharmacological approaches such as diet among both patients and clinicians.

Recent advancements in our understanding of the use of dietary therapies in IBD have grown significantly, making it an essential component of disease management, from prevention to treating active disease to addressing its complications, like malnutrition. This shift is reflected in updated IBD management guidelines, which have transitioned from considering diet as a minor factor to now including diet-specific recommendations [4,5,6]. Notably, exclusive enteral nutrition and the Crohn’s disease exclusion diet (CDED) have proven effective in inducing remission and adherence to a Mediterranean diet is recommended for the maintenance of remission. While emerging evidence supports the potential of dietary therapy to manipulate gut microbiota to restore a healthy profile and mediate intestinal inflammation in IBD patients [7,8], the specific mechanisms by which diet and its components influence intestinal barrier function are poorly understood.

The dysregulation of the intestinal epithelial barrier is one potential mechanism linked to the pathogenesis of IBD. The hypothesis suggests that the dysfunction of the intestinal epithelial barrier and the resulting increase in intestinal permeability (IP) (a measurable functional characteristic of the intestinal epithelial barrier) facilitate the entry of antigens and/or microorganisms into the lamina propria, leading to immune system activation and the initiation or perpetuation of inflammatory responses [9]. While IP has been studied in CD, there is limited knowledge of its role in UC [10,11].

A Western diet—marked by a high consumption of red and processed meats, refined grains, sodium, and processed foods with lower intakes of fruit, vegetables and fermentable fiber—is a recognized risk factor for the development of IBD [12]. Westernized diets are associated with gut microbial dysbiosis, characterized by an increase in colonic mucus-degrading bacteria and a reduction in the production of essential metabolites like short-chain fatty acids, which are vital for maintaining the intestinal barrier [13]. Studies independent of IBD cohorts indicate that lifestyle and dietary factors, such as a Western-style diet, have been correlated with increased IP [14]. On the contrary, a diet rich in polyphenols, fiber, and omega-3 fat are associated with decreased IP [15,16]. Despite these findings, the interaction between diet and IP is poorly understood in IBD. A deeper understanding of these mechanisms is essential for developing personalized diets with the potential to enhance the management and treatment of IBD.

No single biomarker has been identified for the routine assessment of IP. The gold standards, the 51Cr-labeled ethylenediaminetetraacetic acid (51Cr-EDTA) test and the lactulose and mannitol ratio (lac/man ratio) are often impractical for clinical use due to their time-consuming nature, limited availability, and inability to be performed retrospectively [17]. While endogenous proteins have been proposed as biomarkers, no consensus exists on which are reliable for clinical application. Zonulin, a protein that regulates IP by modulating tight junctions between epithelial cells, is upregulated in IBD, suggesting a role in intestinal barrier disruption [18,19]. Lipopolysaccharide binding protein (LBP), an acute-phase protein produced by the liver, binds to bacterial lipopolysaccharides resulting from intestinal translocation [18]. Seethalar et al. (2021) found that LBP is a more robust marker for IP than zonulin, with low intraindividual variation and a strong correlation with the lac/man ratio, which was independent of age, BMI, and sex [18]. In IBD, LBP levels are relevant for monitoring disease activity and assessing therapeutic response, though its use has to date been limited to research settings [18,20,21].

In the current study, we examined serum zonulin and LBP concentrations and their correlation with clinical disease activity indices and dietary factors, including macronutrients, micronutrients, and dietary quality measures in patients with UC.

## 2. Material and Methods

### Subjects and Protocols

This is a cross-sectional analysis of baseline serum samples and associated metadata from a previous dietary intervention study (reported elsewhere) [22]. Thirty-seven participants with UC with complete clinical, biological and diet data sets were analyzed. The trial was registered at clinicaltrials.gov as NCT04474561 and was approved by the University of Calgary Conjoint Health Research Ethics Board (REB16-2491).

## 3. Measures

### 3.1. Biomarker Analyses

Peripheral venous blood samples were collected into SST tubes by a trained phlebotomist. After the collection of whole blood, the sample was allowed to clot at room temperature for 30 min, centrifuged at 1500× *g* for 10 min in refrigerated centrifuge (4 °C) serum removed and frozen at −80 °C until analysis. Both serum zonulin and lipopolysaccharide-binding protein (LBP) levels were measured via an ELISA kit (Elabscience, Catalog Number: E-EL-H5560 and Invitrogen (Waltham, MA, USA), Catalog Number: EH297RB, respectively).

### 3.2. Dietary Analyses

Diet assessments and analyses were conducted as previously described [22]. In brief, food and beverage intake were assessed using two non-consecutive 24 h food recalls via the Automated Self-Administered 24 h (ASA-24^®^) Dietary Assessment Tool (Canadian version) [23]. Portion sizes and nutrient data were downloaded from the ASA-24 researcher website and compared with the dietary reference intakes (DRIs) to ensure both macronutrient and micronutrient adequacy [24]. Diet quality was measured using the Healthy Eating Index-2020 (HEI-2020) [25]. The total HEI is composed of two parts: adequacy components, which are foods recommended for consumption to promote health and moderation components (moderation HEI), and are foods that should be limited. Additionally, the English version of the PREvencion con DIetaMEDiterranea 14-item Mediterranean Diet Adherence Screener was used to calculate a Mediterranean diet score (MEDAS) [26], as previously described [22].

### 3.3. Statistical Analysis

The normality of variable distribution was assessed with the Kolmogorov–Smirnov test. Spearman’s correlation coefficient was used to evaluate the correlation between variables. Continuous variables were compared using the Student’s *t*-test for normally distributed data or the Mann–Whitney test for non-normal distributions. Categorical variables were analyzed with the Chi-square test or Fisher’s exact test, as appropriate. All statistical analyses were conducted using GraphPad Prism, version 10.2.3 (San Diego, CA, USA). A *p*-value < 0.05 was considered statistically significant.

## 4. Results

### 4.1. Clinical Characteristics

The participants had a median age of 35 years (range 29–47 years), male (*n* = 20, 54%), and an average BMI of 26, indicating a higher-than-normal BMI. Most participants (76%, *n* = 27) were in clinical remission or had mild disease activity (a PMS of 4 or less). Complete demographics are provided in Table 1.

### 4.2. Dietary Characteristics

Table 2 presents the adjusted nutrient intakes per 1000 kcal for the participants. The analysis of the micronutrients revealed that the participants did not meet the recommended dietary allowance or adequate intake levels for fiber, calcium, folate, vitamin E, vitamin D, and choline. We also assessed the dietary patterns of the exploratory cohort using the Healthy Eating Index (HEI) and the Mediterranean Diet Score (MEDAS). The median HEI score was a median 69 (interquartile range (IQR): 63–76), which exceeds the average score of 58.8 for Canadians aged two or older [27]. Participants had a median MEDAS score of 5 (IQR: 4–6), which is considered weak adherence compared to a healthy group of individuals [28].

### 4.3. Correlations Between Serum Zonulin with Dietary Characteristics

Figure 1A shows the Spearman rank correlations between serum zonulin and various nutritional factors. Both the actual intake (grams/day) and adjusted intake (per 1000 kcal) of niacin (r = −0.40, *p* = 0.02) and vitamin B12 (r = −0.40, *p* = 0.02) were negatively associated with serum zonulin. Adjusted protein intake (r = −0.39, *p* = 0.02) and olive oil consumption (r = −0.38, *p* = 0.02) were also negatively associated with serum zonulin. There was a trend towards significance for serum zonulin and BMI (r = 0.32, *p* = 0.06). The non-significant correlations are shown in Appendix A.

### 4.4. Correlations Between Serum LBP with Dietary Characteristics

LBP was found to be positively correlated with the total fat grams (r = 0.42, *p* = 0.01), vitamin B2-mg (r = 0.37, *p* = 0.03), refined grains (r = 0.35, *p* = 0.04), and monounsaturated fat-grams (r = 0.35, *p* = 0.04). Conversely, negative correlations were observed with fruit consumption (e.g., bananas, grapes, peaches, plums, cherries, mangoes, pineapple, apricots, kiwi, and papayas, as categorized by the ASA24 software (r = −0.50, *p* = 0.002), total fruit and vegetable intake (r = −0.41, *p* = 0.02), vitamin C (r = −0.37, *p* = 0.03), moderation HEI (r = −0.40, *p* = 0.02), and total HEI (r = −0.33, *p* = 0.05), where a higher moderation HEI (less saturated fat, added sugar, and sodium) and total HEI indicated better diet quality.

### 4.5. Correlations Between Disease Activity, Medications, and Markers of Intestinal Permeability

Serum zonulin did not correlate with FCP, PMS, or IBD-related medications (Appendix A), whereas there was a positive correlation between LBP and PMS (r = 0.40, *p* = 0.03), intake of antibiotics (r = 0.52, *p* = 0.00), anti-TNF medications (r = 0.42, *p* = 0.02), and a marginally significant trend for FCP (r = 0.31, *p* = 0.06) (Figure 1B).

## 5. Discussion

This cross-sectional analysis investigates the associations between IP, dietary components, and disease activity in UC, marking the first study to examine the role of diet and its impact on IP markers, specifically lipopolysaccharide-binding protein (LBP) and zonulin, in UC patients.

Diet has emerged as a key non-pharmacological target to reduce disease activity and symptom burden in IBD, yet there is limited evidence on how diet affects IP in UC. The microbiome plays a pivotal role in maintaining intestinal health, with health-associated microbiota regulating key aspects of epithelial barrier function, such as tight junction integrity, angiogenesis, vascular permeability, and immune regulation [29]. Diet significantly influences gut microbiota composition, impacting these processes. Nutrients modulate tight junction proteins (e.g., claudins, occludin, and zonula occludens), with fiber-rich diets promoting short-chain fatty acid (SCFA) production, strengthening the intestinal barrier [30]. Conversely, Western diets—which are low in fiber and high in processed foods—can induce dysbiosis, inflammation, and increased permeability [29]. Nutrients such as SCFAs, zinc, and vitamin D support epithelial repair and fiber enhances mucus layer integrity, whereas processed foods and oxidative stress can damage the barrier [30]. Despite the potential role of diet in managing IP and UC, more research is needed to understand these relationships [29]. This study found that LBP was negatively correlated with whole fruits, vegetables, and vitamin C intake. Although the relationship between these dietary components is not well understood in IBD, previous studies have shown similar negative correlations in the general population [31,32]. In a previous study of patients with IBD, the regular consumption of fruits and vegetables was associated with a 44% reduction in the prevalence of disease activity compared to those who do not eat these foods [33]. Fruits and vegetables are rich in macro-accessible carbohydrates, micronutrients, flavonoids, and other bioactive compounds that possess anti-inflammatory and antioxidant properties which support the integrity of the epithelial barrier and can influence the composition and function of the microbiota, potentially benefiting intestinal inflammation [34]. Our analysis also found that higher dietary quality indices, such as total HEI and moderation HEI, were negatively associated with LBP levels. This aligns with general population studies linking LBP levels with the adherence to a Mediterranean diet [35]. Our cohort did not demonstrate this relationship, possibly due to limitations of the Mediterranean Diet Adherence Screener (MEDAS), which uses a binary classification system and oversimplifies dietary patterns [4,35].

This study confirmed a positive correlation between LBP and the consumption of total fat and refined grains, which are both components of a Western diet, consistent with previous research shown in healthy populations [36]. Although high-fat diets have been shown to increase serum LBP, the specific impact of different fat sub-types on LBP remains unclear. A recent systematic review concluded that meals rich in saturated fatty acids increase plasma LBP concentrations, while meals rich in polyunsaturated fatty acids reduce LBP concentrations [37]. However, the effect of monounsaturated fatty acid (MUFA)-rich diets could not be conclusively determined due to conflicting study results. Interestingly, our results showed a positive correlation between monounsaturated fatty acids (MUFAs) and LBP, a finding that requires further investigation.

A weak positive correlation was observed between serum zonulin and BMI. In other IBD cohorts, increases in BMI and zonulin levels have been shown to increase in parallel, contributing to the development of a low-grade inflammatory state, further exacerbating inflammation in this population [19,38].

Interestingly, we found no correlation between zonulin levels and dietary quality indices (i.e., MEDAS and HEI), which contrasts with studies in healthy individuals where zonulin has been positively correlated with several other dietary factors, including the total calories’ total protein, total carbohydrates, and fatty acid composition of the diet [15,16] Whereas we found that dietary factors such as vitamin B12, niacin, protein intake, and olive oil consumption were negatively correlated with zonulin, our results are contrary to the results found in healthy individuals where zonulin has been positively correlated with protein intake and vitamin B12 in healthy adult women [15]. The discrepancy between the results may be explained by the type of protein consumed (plant-based vs. animal protein), as studies have shown healthy individuals following a vegan diet for more than three years have significantly lower zonulin levels compared to those on an omnivore diet [39]. These findings suggest that further research is needed to understand how nutrients influence zonulin levels in IBD patients.

We observed a significant positive association between serum LBP and measures of disease activity, including PMS (r = 0.40, *p* < 0.05) and a trend towards significance with FCP (r = 0.31, *p* = 0.06). Two recent studies demonstrated that LBP is correlated with FCP (r = 0.41, *p* < 0.0001 and r = 0.42, *p* < 0.01) for evaluating endoscopic activity in both UC and CD [21,40]. LBP does not appear to be influenced by steroids, and is independent of age, sex, and BMI [40]. Additionally, it shows very low intraindividual variation over time in individuals ranging from normal weight to obese [18]. This suggests the robustness of LBP as a biomarker for IP and disease activity. In contrast, serum zonulin did not correlate with disease activity, a finding that aligns with the results from other studies [38] and raises questions about the reliability of zonulin as a marker of IP in IBD.

Lastly, we identified inadequate intakes of fiber, calcium, folate, vitamin D, vitamin E, and iron (in females), consistent with the well-documented deficiencies in patients with IBD, suggesting that inadequate intakes in this population continue to be of concern [41,42,43].

Despite several limitations, including a small sample size, potential recall bias, and a cross-sectional design that restricts causal inferences, this study lays the groundwork for future research. The retrospective nature of the design limited our ability to collect additional data, such as endoscopy results and disease location. Further investigation is needed to determine whether LBP is sensitive to dietary changes, disease location, and disease activity in both UC and IBD. Controlled dietary interventions and longitudinal studies are essential to validate these findings and better understand the role of intestinal permeability (IP) in IBD management. This research sets the stage for a more comprehensive exploration of the value and implications of IP in IBD.

## 6. Conclusions

In summary, LBP levels correlated with the baseline disease activity, measured by PMS, whereas zonulin did not. We observed negative correlations between zonulin and B vitamins (niacin and vitamin B12), as well as between LBP and healthy diet parameters (fruit intake, HEI, vitamin C). These findings suggest that LBP may be a potential biomarker for disease activity and dietary influences on IP in IBD, particularly in UC. In line with current recommendations, increased fruit and vegetable intake (with textural modifications as necessary) is recommended as a component of a healthy eating pattern for all patients with IBD with this study providing potential mechanistic insights [4]. These results should be validated in further randomized controlled trials in IBD.

## Figures and Tables

**Figure 1 diagnostics-14-02629-f001:**
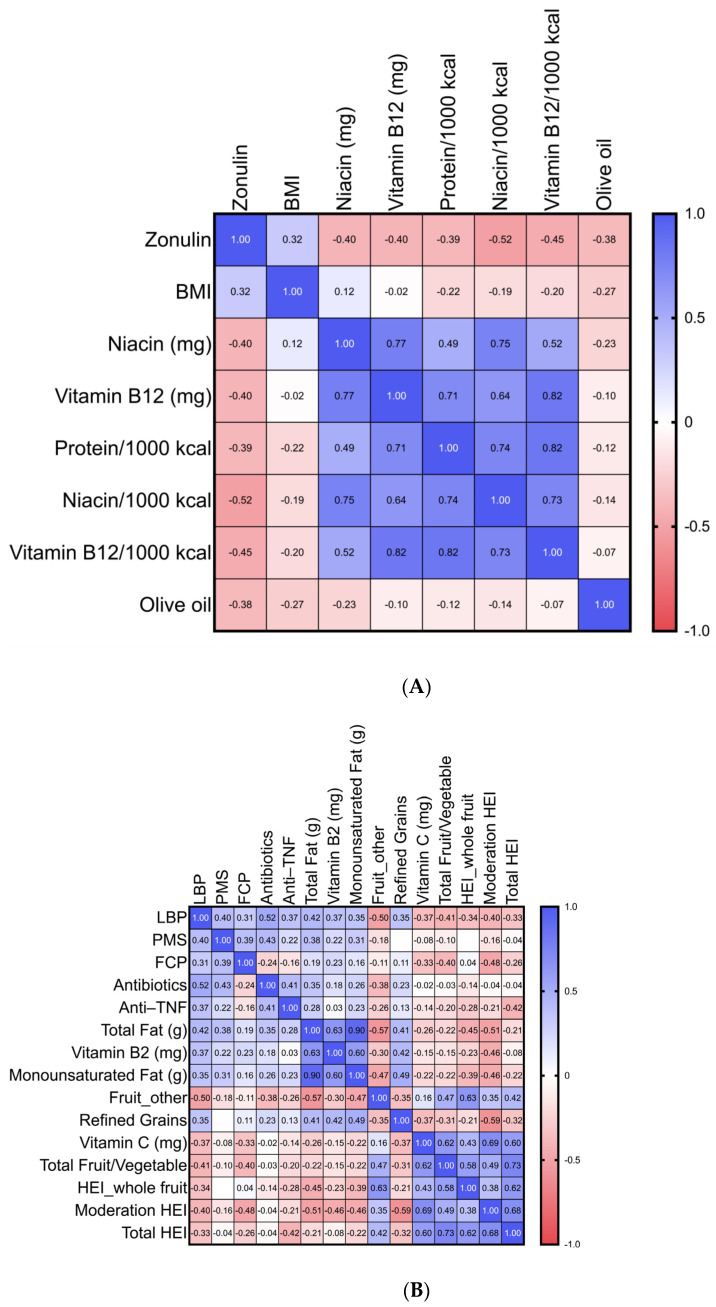
(**A**). Heatmap of the dietary characteristics and their association with serum zonulin. The heatmap illustrates the distribution of various dietary characteristics that correlate with serum zonulin levels. Each row represents a different component. The color gradient indicates the strength and direction of the correlation, with darker blue shades representing a positive correlation and lighter shades of red representing a negative correlation. Only *p* values less than 0.05 are shown. The non-significant correlations are shown in Appendix A. Abbreviations: BMI, body mass index; mg, milligrams (actual grams consumed per day);/1000 kcal refers to nutrient intake adjusted per 1000 kcal. (**B**). Heatmap of dietary characteristics and their association with serum lipopolysaccharide-binding protein (LBP). The heatmap illustrates the distribution of various dietary characteristics that significantly correlate with serum LBP levels. Each row represents a different component. The color gradient indicates the strength and direction of the correlation, with darker blue shades representing a positive correlation and lighter shades of red representing a negative correlation. Only *p* values of less than 0.05 are shown. The non-significant correlations are shown in Appendix A. Refined grains are grains that do not contain all the components of the entire grain kernel (oz. eq.) as defined by the ASA24 software (https://epi.grants.cancer.gov/asa24/, accessed on 31 July 2021) [23]. Abbreviations: PMS, partial Mayo score; FCP, fecal calprotectin; Anti-TNF, anti-tumor necrosis factor; HEI, Healthy Eating Index; g, grams; mg, milligrams.

**Table 1 diagnostics-14-02629-t001:** Characteristics of the UC cohort.

Patient Characteristics	Baseline (*n* = 37)
Age (years)	35 (29–47)
Sex (male, %)	20 (54)
BMI (kg/m^2^)	26 (24–28) (*n* = 35)
**Disease activity**	
Partial Mayo score	3 (1–5)
Fecal calprotectin (ug/g)	107 (53–2441)
**Treatments ^a^**	
5-ASA (%)	20 (50.0)
Steroids (%)	12 (30.0)
anti-TNF (%)	8 (20.0)
Immunosuppressants (%)	5 (12.5)
Antibiotics (%)	5 (12.5)
**Intestinal Permeability Biomarkers**	
LBP (ng/mL)	12,570 (9890–21,905)
Zonulin (ug/mL)	141 (66–327)

Abbreviations: LBP, lipopolysaccharide binding protein. Values are presented as median (interquartile range) unless otherwise specified. ^a^ Some patients were taking more than one therapy.

**Table 2 diagnostics-14-02629-t002:** Diet characteristics of the UC cohort (per 1000 kcal per day) (*n* = 37).

	RDA/AI ^a^	Baseline Median (IQR)
Fiber (g)	19 g male 12.5 g female	9.0 (7.0–12)
Calcium (mg)	500 mg	368 (320–604)
Iron (mg)	4 mg male 9 mg female	6.6 (5.9–7.4)
Magnesium (mg)	210 mg male 160 mg female	162 (132–201)
Phosphorus (mg)	350 mg	736 (584–806)
Potassium (mg)	1700 mg male 1300 mg female	1486 (1141–1680)
Sodium (mg)	750 mg	1614 (1386–20696)
Zinc (mg)	5.4 mg male 4 mg female	5.3 (4.1–7.3)
Copper (mg)	0.45 mg	0.66 (0.50–0.83)
Selenium (mg)	27.5 mg	58 (47–67)
Vitamin C (mg)	45 mg male 37.5 mg female	58 (27–80)
Thiamin (mg)	0.6 mg	0.80 (0.66–0.90)
Riboflavin (mg)	0.65 mg male 0.55 mg female	0.80 (0.66–0.90)
Niacin (mg)	8 mg male 7 mg female	11 (7.6–14)
Vitamin B6 (mg)	0.65 mg male 0.55 mg female	0.84 (0.68–1.3)
Folate Total (DFE)	200 DFE	176 (139–201)
Vitamin B12 (mg)	1.2 mg	1.8 (3.5–2.5)
Vitamin A (RAE)	450 RAE male 350 RAE female	356 (254–545)
Vitamin E, alpha-tocopherol (mg)	7.5 mg	4.4 (3.5–6.0)
Vitamin K, phylloquinone (mcg)	60 mcg male 45 mcg female	54 (33–78)
Vitamin D (mcg)	7.5 mcg	1.8 (0.89–2.9)
Choline (mg)	225 mg male 193 mg female	165 (111–205)
Diet Quality Scores		
MEDAS (total)	5 (4–6)	
HEI (total)	69 (63–76)	
Moderation HEI	23 (20–29)	

Abbreviations: RDA/AI, recommended dietary allowance/adequate intake; mg, milligram; mcg, microgram; DFE, dietary folate equivalent; RAE, retinol activity equivalents; MEDAS, Mediterranean Diet Adherence Score; HEI, Healthy Eating Index, ^a^ estimated recommended adjusted intake to be 50% of RDA/AI. Values are presented as median (interquartile range).

## Data Availability

The data underlying this article are available in the article and in the online Appendix A.

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
