# Peer review of "Biomarkers of Intestinal Permeability Are Influenced by Diet in Patients with Ulcerative Colitis—An Exploratory Study"

_diagnostics, 2024, doi:10.3390/diagnostics14232629_

Round 1

Reviewer 1 Report

Comments and Suggestions for Authors

This study explores the associations between dietary factors, intestinal permeability biomarkers, and disease activity in patients with UC. A very-interesting and hot topic due to increased prevalence of IBD. This study aims to address an underexplored area-the impact of diet on biomarkers related to intestinal permeability in UC. The manuscript is well-writing and the authors explain clearly the methodology of study. 
However, there is some concerrns. Please make the following revisions.
1. Please include data on the location of ulcerative colitis (e.g., proctitis, left-sided colitis, pancolitis) and analyze whether the disease location affects intestinal permeability. 
2. Analyze, if the medications affect the permeability. 
3. In the discussion section, please describe the limitations of the study: small sample, no current endoscopy, small recall period for the dietary assessment tool. 
4. In the discussion section, analyze potential mechanistic pathways that could explain the link between diet, intestinal permeability, and UC disease activity.
5. Provide suggestions for future research, such as studies including Crohn's disease patients, controlled dietary intervention trials, and longitudinal studies.

Author Response

 Biomarkers of Intestinal Permeability Are Influenced by Diet in Patients with Ulcerative Colitis – An Exploratory Study

*Natasha Haskey, 1,2 *Max Eisele 1, Andreina Bruno 3, Raylene A. Reimer 4, Munazza Yousuf 1, Lorian M. Taylor 1, Remo Panaccione 1 and Subrata Ghosh, 5 Maitreyi Raman 1

Overall:

We would like to thank both the reviewers and the board of editors for their valuable suggestions and the opportunity to provide a revision. Considering the new information provided, we believe the manuscript is stronger and more comprehensive shedding additional light on the association between markers of disease activity (fecal calprotectin and partial Mayo score) and dietary factors in patients with UC. Below we address the reviewer’s comments.

Comments from the Editors and Reviewers:

Reviewer 1:

This study explores the associations between dietary factors, intestinal permeability biomarkers, and disease activity in patients with UC. A very-interesting and hot topic due to increased prevalence of IBD. This study aims to address an underexplored area-the impact of diet on biomarkers related to intestinal permeability in UC. The manuscript is well-writing and the authors explain clearly the methodology of study. However, there is some concerns. Please make the following revisions.

  1. Please include data on the location of ulcerative colitis (e.g., proctitis, left-sided colitis, pancolitis) and analyze whether the disease location affects intestinal permeability.

We thank the reviewer for their insightful comment regarding the influence of disease location on intestinal permeability in ulcerative colitis. Unfortunately, our current dataset did not include stratified information on UC subtypes by location (proctitis, left-sided colitis, pancolitis). We agree that investigating the link between UC location and permeability could be impactful. We plan to include such stratification in future studies with larger cohorts to better address this relationship.

We have added the following to the discussion:

“Despite several limitations, including a small sample size, potential recall bias, and a cross-sectional design that restricts causal inferences, this study lays the groundwork for future research. The retrospective nature of the design limited our ability to collect additional data, such as endoscopy results and disease location. Further investigation is needed to determine whether LBP is sensitive to dietary changes, disease location, and disease activity in both UC and IBD. Controlled dietary interventions and longitudinal studies are essential to validate these findings and better understand the role of intestinal permeability (IP) in IBD management. This research sets the stage for a more comprehensive exploration of the value and implications of IP in IBD.”

  1. Analyze, if the medications affect the permeability.

Thank you for the suggestion. We did run the stats on medications and they are included in our Supplementary Table 1. Medications did not influence IP when measured via zonulin. We did find IP was influenced by antibiotics and anti-TNF medications when measured by LBP.

As a result, we changed the wording in section 4.5 to include your suggestion (LINE 182):

4.5. Correlations between Disease Activity, Medications and Markers of Intestinal Permeability

“Serum zonulin did not correlate with FCP, PMS or IBD-related medications (Supplementary Table S1), whereas there was a positive correlation between LBP and PMS (r=0.40, p=0.03), intake of antibiotics (r=0.52, p=0.00), anti-TNF medications (r=0.42, p=0.02) and a marginally significant trend for FCP (r=0.31, p=0.06) (Figure 1B).”

We have also amended Figure 1B to include the anti-TNF.

  1. In the discussion section, please describe the limitations of the study: small sample, no current endoscopy, small recall period for the dietary assessment tool.

Thank you for the suggestion. We have added the following to the discussion:

“Despite several limitations, including a small sample size, potential recall bias, and a cross-sectional design that restricts causal inferences, this study lays the groundwork for future research. The retrospective nature of the design limited our ability to collect additional data, such as endoscopy results and disease location. Further investigation is needed to determine whether LBP is sensitive to dietary changes, disease location, and disease activity in both UC and IBD. Controlled dietary interventions and longitudinal studies are essential to validate these findings and better understand the role of intestinal permeability (IP) in IBD management. This research sets the stage for a more comprehensive exploration of the value and implications of IP in IBD.”

  1. In the discussion section, analyze potential mechanistic pathways that could explain the link between diet, intestinal permeability, and UC disease activity.

Thank you for the suggestion, we have added the following to the discussion:

“Diet has emerged as a key non-pharmacological target to reduce disease activity and symptom burden in IBD, yet there is limited evidence on how diet affects IP in UC. The microbiome plays a pivotal role in maintaining intestinal health, with health-associated microbiota regulating key aspects of epithelial barrier function, such as tight junction integrity, angiogenesis, vascular permeability, and immune regulation.29 Diet significantly influences gut microbiota composition, impacting these processes. Nutrients modulate tight junction proteins (e.g., claudins, occludin, and zonula occludens), with fiber-rich diets promoting short-chain fatty acid (SCFA) production, strengthening the intestinal barrier.30 Conversely, Western diets—low in fiber and high in processed foods—can induce dysbiosis, inflammation, and increased permeability.29 Nutrients such as SCFAs, zinc, and vitamin D support epithelial repair and fiber enhances mucus layer integrity, wheras processed foods and oxidative stress can damage the barrier.30 Despite the potential role of diet in managing IP and UC, more research is needed to understand these relationships.”

  1. Provide suggestions for future research, such as studies including Crohn's disease patients, controlled dietary intervention trials, and longitudinal studies.

Thank you for the suggestion. We have added the following statement to the discussion

“Despite several limitations, including a small sample size, potential recall bias, and a cross-sectional design that restricts causal inferences, this study lays the groundwork for future research. The retrospective nature of the design limited our ability to collect additional data, such as endoscopy results and disease location. Further investigation is needed to determine whether LBP is sensitive to dietary changes, disease location, and disease activity in both UC and IBD. Controlled dietary interventions and longitudinal studies are essential to validate these findings and better understand the role of intestinal permeability (IP) in IBD management. This research sets the stage for a more comprehensive exploration of the value and implications of IP in IBD.”

Reviewer 2 Report

Comments and Suggestions for Authors

an interesting starting point

clearly as stated in the conclusions the data must be developed in a larger study with control group

Author Response

Reviewer 2:

Clearly, as stated in the conclusions the data must be developed in a larger study with control group

Thank you for the suggestion and we hope to build upon this exploratory study in our future research cohorts.

Reviewer 3 Report

Comments and Suggestions for Authors

The current study examined the effect of dietary macro- and micronutrients on zonulin and lipopolysaccharide binding protein, as biomarkers of intestinal permeability in patients with ulcerative colitis. There are some points that should be taken into consideration while reviewing the manuscript:

The clinical trial number written at the end of the abstract part. Please either add "Trial information:" sentence before it or remove it and just added it in the section related to "Ethics Statement" at the end of the manuscript.

There is no need to repeat the abbreviations, for example line 80, intestinal permeability (IP.(

Line 106, “Separated into serum and plasma.” How is blood separated into serum and plasma? Blood is separated into serum or plasma based on the presence or absence of an anticoagulant.

It is well known that intestinal permeability is linked to many disease including hepatic diseases, diabetes, and irritable bowel syndrome. it is known also that zonulin and lipopolysaccharide binding protein are biomarker for intestinal permeability. The discussion part must be re-edited because it is merely a repetition of the results part and there is no explanation of the effect of food and its components, especially since there are published studies on the effect of nutrients and green leafy plants on zonulin. Theses studies can be used to write a strong discussion part.

In the conculsion patr, line 270, "These findings suggest that LBP may be a potential biomarker for disease activity and dietary components in IBD, particularly in UC”, a very confusing sentence: is LBP a biomarker for dietary components ? Please rewrite the conclusion part. Did food components affect the biomarker of intestinal permeability in patients with ulcerative colitis? What are the most influential macro- and micronutrients? And what is recommended to patients with ulcerative colon from a nutritional point of view based on the results ? .

Author Response

Reviewer 3:

The current study examined the effect of dietary macro- and micronutrients on zonulin and lipopolysaccharide binding protein, as biomarkers of intestinal permeability in patients with ulcerative colitis. There are some points that should be taken into consideration while reviewing the manuscript :

The clinical trial number written at the end of the abstract part. Please either add "Trial information:" sentence before it or remove it and just added it in the section related to "Ethics Statement" at the end of the manuscript.

Removed from abstract (Line 36) and is included in the ethics statement at the end of the manuscript.

There is no need to repeat the abbreviations, for example line 80, intestinal permeability (IP (.

Thank you for your comment – we have corrected this within the manuscript.

Line 106, “Separated into serum and plasma.” How is blood separated into serum and plasma? Blood is separated into serum or plasma based on the presence or absence of an anticoagulant .

This section was rewritten in the manuscript to provide clarification:

“Peripheral venous blood samples were collected into SST tubes by a trained phlebotomist. After collection of the whole blood, the sample was allowed to clot at room temperature for 30 minutes, centrifuged at 1500 x g for 10 minutes in a refrigerated centrifuge (4C), serum was removed, and frozen at –80◦C until analysis.”

It is well known that intestinal permeability is linked to many disease including hepatic diseases, diabetes, and irritable bowel syndrome. it is also known that zonulin and lipopolysaccharide-binding proteins are a biomarkers for intestinal permeability . The discussion part must be re-edited because it is merely a repetition of the results part and there is no explanation of the effect of food and its components, especially since there are published studies on the effect of nutrients and green leafy plants on zonulin. These studies can be used to write a strong discussion part.

Thank you for your suggestion – we have edited the discussion to add more information on food components and how they influence IP.

In the conclusion part, line 270, "These findings suggest that LBP may be a potential biomarker for disease activity and dietary components in IBD, particularly in UC,” a very confusing sentence: is LBP a biomarker for dietary components ? Please rewrite the conclusion part. Did food components affect the biomarker of intestinal permeability in patients with ulcerative colitis? What are the most influential macro- and micronutrients? And what is recommended to patients with ulcerative colon from a nutritional point of view based on the results?

Thank you for the suggestion, we have rewritten the conclusion:

In summary, LBP levels correlated with baseline disease activity, measured by PMS, whereas zonulin did not. We observed negative correlations between zonulin and B vitamins (niacin and vitamin B12), as well as between LBP and healthy diet parameters (fruit intake, HEI, vitamin C). These findings suggest that LBP may be a potential biomarker for disease

activity and dietary influences on IP in IBD, particularly in UC. In line with current recommendations, increased fruit and vegetable intake (with textural modifications as necessary) is recommended as a component of a healthy eating pattern for all patients with IBD with this study providing potential mechanistic insights.4 These results should be validated in further randomized controlled trials in IBD.

Round 2

Reviewer 3 Report

Comments and Suggestions for Authors

All the suggested amendments were done and the inquiries were answered. Thanks for the authors.